# Examining the association between fetal *HLA-C,* maternal *KIR* haplotypes and birth weight

Caitlin Stephanie Decina[1,2]*, Nicole M. Warrington[2,3], Robin N. Beaumont[1], Beilei Bian[2],
Caroline Brito Nunes[2], Geng Wang[2], William L. Lowe Jr[4], David Squire[5,6],
Damjan Vukcevic[5,6,7], Stephen Leslie[5], Rachel M. Freathy[1‡], David M. Evans[2,3,8‡]

1 Department of Clinical and Biomedical Sciences, Faculty of Health and Life Sciences, University of
Exeter, Exeter, United Kingdom, 2 Centre for Population and Disease Genomics, Institute for Molecular
Bioscience, University of Queensland, Brisbane, Queensland, Australia, 3 Frazer Institute, University of
Queensland, Brisbane, Queensland, Australia, 4 Department of Medicine, Feinberg School of Medicine,
Northwestern University, Chicago, Illinois, United States of America, 5 Department of Mathematics and
Statistics, School of Mathematics and Statistics, University of Melbourne, Melbourne, Victoria, Australia,
6 Melbourne Integrative Genomics, University of Melbourne, Melbourne, Victoria, Australia, 7 Department
of Econometrics and Business Statistics, Monash University, Melbourne, Australia, 8 Medical Research
Council Integrative Epidemiology Unit, University of Bristol, Bristol, United Kingdom

‡ These authors jointly directed to this work.
* cd629@exeter.ac.uk

doi.org/10.1371/journal.pgen.1012102

Institute for Biotechnology, UNITED STATES OF
AMERICA

**Peer Review History:** PLOS recognizes the
benefits of transparency in the peer review
process; therefore, we enable the publication
of all of the content of peer review and
author responses alongside final, published
articles. The editorial history of this article is
available here: https://doi.org/10.1371/journal.
pgen.1012102

## Abstract

Human birth weight is under stabilizing selection, seeking balance between extremes
of high and low, thereby reducing fetal and maternal perinatal mortality risk. Certain
combinations of maternal killer immunoglobulin-like receptor (*KIR*) and paternally
derived fetal human leuokocyte antigen-C (*HLA-C*) alleles were previously associated with higher risk of high and low birth weight in a study with limited sample size
(n = 1,316). Using recently developed methods to impute *HLA* and *KIR* haplotypes
using single nucleotide polymorphism (SNP) genotype data, we tested associations
of fetal *HLA* and maternal *KIR* genotypes with offspring birth weight in a large sample. We imputed *KIR* haplotypes using the KIR*IMP imputation software in 10,602
mother-offspring pairs of European descent from singleton pregnancies from five
studies. Using mixed linear regression models to account for mothers with multiple
children, we tested associations between maternal *KIR A* vs *B* haplotypes (*AA, AB/
BA, BB* genotypes) as well as copy number of activating receptor gene *KIR2DS1* (0,
1, 2 copies of the gene) in the presence of fetal *HLA C1/C2* alleles, and offspring birth
weight. Associations were analyzed in each cohort before performing a meta-analysis
to estimate the interaction effects between maternal *KIR* and fetal *HLA-C2* on birth
weight across the entire sample. The *KIR* haplotypes achieved imputation accuracy
estimated at >95% in most of the cohorts. No interaction effects were observed
between either the maternal *A* vs. *B* haplotype or the maternal *KIR2DS1* locus and
fetal *HLA-C*. When specifically trying to replicate the previously associated combination of maternal *KIR2DS1* and paternally inherited fetal *HLA-C2*, there was a negligible change in offspring birth weight for each additional *KIR2DS1* allele and *HLA-C2* of

**Data availability statement:** The data that underlie the reported findings of this study consist of sensitive human research participant data. It is held securely to protect the interests of research participants in line with the ethics committee policies and guidance for each source study, and as such is not publicly available. However, the data analyzed in this study are available to researchers through open collaboration. Interested parties may request or apply for access from the relevant organizations. The genetic and phenotype datasets generated by UK Biobank used in the current study, accessed under application number 53641, are available via the UK Biobank data access process (see http://www.ukbiobank.ac.uk/register-apply/). Detailed information about the genetic data available from UK Biobank is available at http://www.ukbiobank.ac.uk/scientists-3/genetic-data/ and http://biobank.ctsu.ox.ac.uk/crystal/label.cgi?id=100314. Researchers interested in accessing the EFSOCH data should email the Exeter Clinical Research Facility at crf@exeter.ac.uk. The HAPO genotype and accompanying phenotype data on European mothers and offspring used in this study, along with additional data for participants of Mexican American, Thai, and Afro-Caribbean ancestry are available through dbGaP (https://www.ncbi.nlm.nih.gov/projects/gap/cgi-bin/study.cgi?study_id=phs000096.v4.p1). The ALSPAC data management plan describes in detail the policy regarding data sharing, which is through a system of managed open access. The data used in this study are linked to ALSPAC project number B2388. To request access to the data included in this paper and all other existing ALSPAC data: (i) Please read the ALSPAC access policy, which describes the process of accessing the data and samples in detail and outlines the costs associated with doing so, (ii) you may also find it useful to browse the fully searchable ALSPAC research proposals database, which lists all research projects that have been approved since April 2011, and (iii) please submit your research proposal for consideration by the ALSPAC Executive Committee. You will receive a response within 10 working days to advise you whether your proposal has been approved. If you have any questions about accessing data, please email alspac-data@bristol.ac.uk. Please note that the study website contains

paternal origin (7g lower birth weight per allele [95% CI: -54, 40], $P = 0.78$). We found little evidence of association between birth weight and maternal *KIR* haplotypes or fetal *HLA-C2* and were unable to replicate previously reported findings. Our observations reinforce the importance of replication and the use of large sample sizes in the validation of genetic associations.

## Author summary

Babies born with very high or low birth weights and their mothers are at a higher risk of illness and death than babies with weights close to average. Genes involved in the maternal immune system, called "*KIR*", and the fetal immune system, called "*HLA-C*", are important for early development of the placenta. Previously published research using a small sample has provided evidence for the role of interacting combinations of these genes in driving the spectrum of birth weight and maintaining the balancing selection of mother-child physiology that results in healthy birth outcomes. Here we harness recently developed methods to impute these genetic data to test associations of maternal *KIR* and fetal *HLA* with child's birth weight in a larger sample. By examining >10,000 European ancestry mother-child pairs, we found no relationship between child's birth weight and any of the genetic combinations we tested of *KIR* in the mother and *HLA-C* in the fetus. We show that despite biological plausibility, it is important to validate genetic associations through replication and using the largest sample sizes possible. Future research could benefit from including birth weights in the true extremes of the spectrum, using methods such as high throughput genome sequencing technologies which could provide more accurate data for these gene regions on a larger scale, and investigation in ancestrally diverse populations.

## Introduction

Human birth weight is under stabilizing selection, seeking a balance between extremes of high and low in order to optimize pregnancy outcomes and reduce the risk of fetal and maternal perinatal mortality [1]. Very low birth weight is associated with greater risk of stillbirth and preterm labour, and increased offspring cardiometabolic risk in later life [2–4]. Conversely when babies are born very large, the mother is at risk of experiencing prolonged obstructed labour, which may lead to haemorrhage and/or sepsis. Likewise, the child is at risk of shoulder dystocia and asphyxiation during birth, as well as cardiometabolic morbidities in later life [5,6].

Whilst mother and offspring share 50% of their nuclear genomes, they are antigenically distinct. Thus, invading fetal trophoblast cells will express molecules encoded by paternal genes, identifying the fetus as 'non-self' to the mother. This dynamic brings into play aspects of the immune system, namely the interaction between maternal killer immunoglobulin-like receptors (KIR) expressed on uterine natural

details of all the data that is available through a fully searchable data dictionary and variable search tool: http://www.bristol.ac.uk/alspac/researchers/our-data/. Scientists are encouraged and able to use BiB data. The approval number for this project is SP479. Data requests are made to the BiB executive using the form available from the study website http://www.borninbradford.nhs.uk (please click on 'Science and Research' to access the form). Guidance for researchers and collaborators, the study protocol and the data collection schedule are all available via the website. All requests are carefully considered and accepted where possible.

**Funding:** This work was supported by a PhD studentship granted to C.S.D by the QUEX Institute, a collaborative program between the University of Exeter and the University of Queensland. R.M.F. and R.N.B were supported by a Wellcome Senior Research Fellowship (WT220390). R.M.F. is also supported by a grant from the Eunice Kennedy Shriver National Institute of Child Health & Human Development of the National Institutes of Health under Award Number R01HD101669. N.M.W and G.W. were supported by an Australian National Health and Medical Research Council (NHMRC) Investigator grant (APP2008723). D.M.E. is supported by an NHMRC Investigator grant (APP2017942). The contents of the published material are solely the responsibility of the authors and do not reflect the views of the NHMRC. Genotyping of the EFSOCH study samples was funded by the Wellcome Trust and Royal Society (grant 104150/Z/14/Z). The UK Medical Research Council and Wellcome (Grant ref: 217065/Z/19/Z) and the University of Bristol provide core support for ALSPAC. This publication is the work of the authors and C.S.D, R.M.F. and D.M.E will serve as guarantors for the contents of this paper. A comprehensive list of grants funding (PDF, 330KB) is available on the ALSPAC website. This research was specifically funded by the Wellcome Trust (Grant ref: WT088806). HAPO was supported by grants from the Eunice Kennedy Shriver National Institute of Child Health and Human Development and the National Institute of Diabetes and Digestive and Kidney Diseases (R01-HD34242 and R01-HD34243); the National Center for Research Resources (M01-RR00048 and M01-RR00080); and the American Diabetes Association. Genotyping of the HAPO study samples was also funded by Wellcome Trust

killer cells (uNK) and human leuokocyte antigen-C (HLA-C) present on fetal trophoblast cells [7,8]. The *HLA-C* gene belongs to the HLA class I region (corresponding to major histocompatibility complex [MHC] class I), a highly variable region of the genome that plays a key role in antigen presentation from intracellular sources (including from invading intracellular pathogens), and recognition of one's own cells [9]. HLA-C is the dominant ligand for KIRs in humans and is categorized into two groups: C1 and C2 based on the amino acid present at position 80 [10,11]. HLA-C binding with KIRs regulates uNK function and trophoblast invasion into the uterine wall by producing either positive or negative signals via cytokine secretion depending on whether the KIR possesses activating or inhibitory function [12–15]. *KIR* genotypes can be grouped into two broad haplotypes, *A* and *B*, based on their gene content. The *KIR A* haplotype contains fewer genes which encode mostly inhibitory receptors which bind HLA-C, while the *KIR B* haplotype contains numerous genes encoding mainly activating receptors, although only one, KIR2DS1, can bind with HLA-C [16,17]. Both *HLA-C* and *KIR* genes are highly polymorphic, meaning there are many possible maternal *KIR* and fetal *HLA-C* genetic combinations arising from great numbers of genotypes and considerable haplotype diversity [9,18].

Certain combinations of maternal *KIR* and fetal *HLA-C* alleles have been previously associated with birth weight. Specifically, Hiby et al. 2014 reported that the cytokine-inhibiting maternal *KIR AA* genotype was observed more frequently in low birth weight pregnancies, while the cytokine-activating maternal *KIR BB* genotype was observed more frequently in high birth weight pregnancies compared to average birth weight pregnancies [19]. When investigating the effect of the maternal *KIR2DS1* locus and combinations of fetal *HLA-C2* alleles on birth weight, the authors found an average increase in birth weight of ~250g in pregnancies where the mother possessed *KIR2DS1* and the fetus possessed more *HLA-C2* alleles than their mother. This was in comparison to pregnancies where mothers had 0 copies of *KIR2DS1* and the fetus had fewer or equal numbers of *C2* alleles compared with their mother. It is worth noting that the effect sizes reported in the Hiby et al. study are an order of magnitude greater than the main effect of other common variants that have been robustly associated with birth weight [19–21]. The authors also found that maternal *KIR2DS1* was only associated with increased offspring birth weight when the fetal *C2* allele was paternally inherited compared to maternally inherited [19]. Fetal growth and birth size are dependent on sufficient nutrient delivery via the placental spiral arteries [22,23]. These associations are consistent with KIR and HLA molecule interaction in transformation of the spiral arteries [12], i.e., combinations of fetal *HLA-C2* and maternal activating genotypes (*KIR BB*) may excessively promote arterial transformation, enhancing fetal growth, whilst combinations of fetal *C2* and maternal inhibitory genotypes (*KIR AA*) may insufficiently promote arterial transformation, leading to reduced fetal growth [17].

However, the sample size of the Hiby et al. study was small (n = 1,316 pregnancies), especially compared with recent genome-wide association studies (GWAS) [20,21], and though single nucleotide polymorphisms (SNPs) in the HLA and KIR gene clusters have been identified as associated with birth weight in GWAS [20,21],

and Royal Society grant 104150/Z/14/Z. BiB data used in this research were funded by the Wellcome Trust (WT101597MA), a joint grant from the UK Medical Research Council (MRC) and UK Economic and Social Science Research Council (ESRC) (MR/N024397/1) and the National Institute for Health Research (NIHR) under its Applied Research Collaboration for Yorkshire and Humber (NIHR200166) and the Clinical Research Network (CRN). This study was supported by the National Institute for Health and Care Research Exeter Biomedical Research Centre. The views expressed are those of the authors and not necessarily those of the NIHR or the Department of Health and Social Care. This research was funded in part, by the Wellcome Trust (Grant number: WT220390). For the purpose of Open Access, the author has applied a CC BY public copyright licence to any Author Accepted Manuscript version arising from this submission. The funders had no role in study design, data collection and analysis, decision to publish, or preparation of the manuscript.

**Competing interests:** The authors have declared that no competing interests exist.

the key interaction findings have not been tested for replication in large independent samples to date. Whilst direct genotyping of *KIR* and *HLA* regions is costly and time intensive, statistical methods are now available to impute *HLA* and *KIR* haplotypes using SNP genotyping data [24–27] and provide an opportunity to investigate *HLA/KIR* interactions in much larger samples.

In this study we used statistical imputation of genome-wide SNP data to generate *HLA* and *KIR* information in five cohorts containing large numbers of mother-offspring pairs (total n = 10,602). Using the estimated *KIR* haplotypes and *HLA-C* genotypes, we tested for association between birth weight and maternal *KIR*/fetal *HLA-C* genetic combinations.

## Methods

### Ethics statement

The UK Biobank has approval from the North West Multi-Centre Research Ethics Committee (MREC) as a Research Tissue Bank (RTB) approval. Participants provided written informed consent.

Ethical approval for the Exeter Family Study of Childhood Health was given by the North and East Devon (UK) Local Research Ethics Committee (approval number 1104), and written informed consent was obtained from the parents of the newborns.

Ethical approval for the study was obtained from the Avon Longitudinal Study of Parents and Children (ALSPAC) Ethics and Law Committee and the Local Research Ethics Committees. Informed consent for the use of data collected via questionnaires and clinics was obtained from participants following the recommendations of the ALSPAC Ethics and Law Committee at the time. Consent for biological samples has been collected in accordance with the Human Tissue Act (2004). ALSPAC used a combination of written informed consent for biological samples and clinical assessments, and implied consent for data collected via returned postal or online questionnaires. Study participants have the right to withdraw their consent for elements of the study or from the study entirely. Full details of the ALSPAC consent procedures are available on the study website (http://www.bristol.ac.uk/alspac/researchers/research-ethics/).

Ethics approval was obtained for the main platform study and all of the individual sub-studies from the Bradford Research Ethics Committee.

### Study populations

**UK Biobank.** The UK Biobank (UKB) is a large prospective cohort study of 500,000 participants from across the UK, aged 40–69 years at baseline, with genetic and phenotypic data collected between 2006 and 2010. The study has collected a wide range of data via various modes of assessment including self-completed questionnaire, interview, physical and functional measures, and sample assays of blood, urine and saliva [28]. Through questionnaire, participants were asked to report their own birth weight and (in the case of mothers) the birth weight of their first child.

We used the raw, version 2 genotype data released by UKB in July 2017. DNA was extracted from stored participant blood samples collected at UKB assessment

centres and genotyped using the UK BiLEVE Axiom Array by Affymetrix and the closely related UKB Axiom Array [29]. Genotyping quality control (QC) and derivation of individuals' genetic ancestry via principal component analysis (PCA) using flashPCA [30] has been described elsewhere [29]. We restricted the sample to mother-offspring pairs according to kinship coefficients estimated by KING software [31], excluding participants who did not pass genotyping QC and those who had elected to withdraw from UKB as of February 2022, leaving 8,498 individuals (4,249 mother-offspring pairs) for analysis. For KIR imputation purposes, the raw genotype data was imputed against the Haplotype Reference Consortium (HRC) v1.1 reference panel [32] in genome build 37 using the Michigan imputation server (https://imputationserver.sph.umich.edu/index.html#!) with Minimac4 software [33,34] and phased using Eagle v2.4 [35].

**Exeter Family Study of Childhood Health (EFSOCH).** The EFSOCH study is a prospective study of children born between 2000 and 2004 and their parents, from a postcode-defined area of Exeter, UK, designed to select a European-ancestry cohort for population homogeneity. Detailed anthropometric measurements were taken and DNA was obtained from fasting blood samples from both parents collected at the study visit at 28 weeks' gestation. Birth measures of children were performed within 12 hours of delivery with birth weight being recorded to the nearest 10 grams. Offspring DNA was obtained from umbilical cord blood at delivery and extracted from spun white cells [36]. Genotyping of EFSOCH samples (n = 969 mothers, 937 fathers and 862 children) was performed using the Illumina Infinium HumanCoreExome-24 array (n = 551,839 SNPs/indels). QC procedures excluded based on DNA sample call rate <98% (n = 50 individuals excluded), SNP call rate <95% (n = 13,151 SNPs excluded), minor allele frequency <1% (n = 257,289 further SNPs excluded), Hardy-Weinberg $P < 1 \times 10^{-6}$ (n = 455 further SNPs excluded), sex mismatch between phenotypic sex and genotypically-derived sex (n = 13 individuals excluded), kinship errors after estimation using King [31] (n = 22 individuals excluded), and PC analysis outliers >4.56 SD from the European cluster mean after analysis to determine ancestry of the sample using flashPCA [37] (n = 21 individuals excluded). This left 2,664 individuals available for analysis who were imputed to the TOPMed r2 reference panel [38] via the TOPMed Imputation Server (https://imputation.biodatacatalyst.nhlbi.nih.gov/) with Minimac4 software [33,34] and phased using Eagle v2.4 [35].

**Hyperglycemia and Adverse Pregnancy Outcome (HAPO) Study.** The HAPO study is an observational study which sought to determine the risks of adverse pregnancy outcomes associated with varying levels of maternal glucose not severe enough to constitute diabetes mellitus. Pregnant women across 15 field centres in nine countries underwent a standard oral glucose tolerance test (OGTT) at ~28 weeks' gestation (between 24 and 32 weeks) [39]. Within 72 hours of delivery, neonatal anthropometrics, including birth weight, were recorded using standardized methods across all centres [40].

At the time of OGTT, blood was collected from participating women from which DNA was isolated, while newborn DNA was isolated from umbilical cord blood samples taken at the time of delivery [41]. A total of 2000 mothers and babies (1000 mother-offspring pairs) included in the current study were genotyped using the Illumina Infinium Global Screening Array. QC was performed, and where individuals with genotype call rate <97.5% (82 samples) were removed and SNPs were removed if they had call rates <98% (2.9% of SNPs) or Hardy-Weinberg equilibrium (HWE) $P < 1 \times 10^{-6}$ (0.05% of SNPs). Genetically determined sex was compared with phenotypically determined sex, excluding samples where conflicts were identified (17 samples). Relationships for mother-offspring pairs were confirmed using KING software [31], applying default King thresholds for kinship, with samples excluded where the expected relationship did not match the reported relationship (14 samples). To derive the genetic ancestry of individuals, principal components (PC) analysis was performed on the genotype data using flashPCA [42]. Visual inspection of PC plots identified outlying individuals for exclusion who did not fit within an ancestry PC cluster (155 samples removed).

Following QC, 1,734 (including 565 complete mother-offspring pairs) individuals and 671,798 SNPs were available for further genetic association analysis. These were imputed using the TOPMed Imputation Server [33,34], phased via Eagle v2.4 [35], and imputed against the TOPMed r2 reference panel [38].

**Avon Longitudinal Study of Parents and Children (ALSPAC).** The Avon Longitudinal Study of Parents and Children (ALSPAC) is a prospective observational cohort study. All pregnant women living in a defined area of Avon, UK (an old

administrative county in the Southwest of England comprising the city of Bristol and surrounding areas) expected to have a live birth between 1st April 1991 and 31st December 1992 were eligible for participation. Initial recruitment gave rise to 14,541 pregnancies of which there were 14,676 known fetuses, resulting in 14,062 live-born children. Among those live births, 13,988 were alive at 1 year of age [43,44]. Phenotypic data for both mothers and children were extracted from medical records, where child's birth weight was obtained (in grams) from obstetric clinical records. Repeated blood sampling at routine antenatal clinical visits and at follow-up clinics for mothers and children allowed for creation of a DNA bank including 11,343 children and 10,321 mothers [43,44]. Maternal genome-wide SNP data were obtained using the Illumina 660 Quad Array, and children's data were genotyped using the Illumina 550 Quad Array. QC involved exclusion based on MAF < 1%, HWE < $1 \times 10^{-6}$, sex mismatch, kinship errors, and 4.56 SD from the cluster mean of any sub-populations cluster [45]. Ancestry PCs were generated in the 1000 Genomes sample in order to classify individuals according to the 3 main 1000 genomes populations (European, African, South Asian) and separate out those of European genetic similarity [45]. Following QC, genotype data were imputed against the HRC v1.1 reference panel using the Michigan imputation server [33,34] and phased via Eagle v2.4 [35], giving rise to 17,827 individuals of mainly genetically defined European ancestry with genetic data for further analysis.

**Born in Bradford (BiB).** Born in Bradford (BiB) is a multi-ethnic cohort (largely bi-ethnic with populations of Pakistani and White British individuals) study established in 2007 in Bradford, UK. Women targeted for recruitment were all those booked in for delivery at the Bradford Royal Infirmary and offered an OGTT at approximately 28 weeks' gestation. Between March 2007 and November 2010, >80% of women who attended the OGTT enrolled in the study (12,453 women with 13,776 pregnancies). At recruitment, women completed a comprehensive interviewer-administered questionnaire, were weighed and measured, and had blood and urine samples taken. At birth, infants' anthropometry was assessed, and umbilical cord samples were collected. From these maternal and infant blood samples, DNA was extracted for approximately 10,000 mother-offspring pairs [46,47]. Maternal and fetal samples were genotyped using two separate chips, an Illumina HumanCoreExome (CE) array and Illumina Infinium Global Screening array (GSA) [48]. Individuals were excluded based on mismatch between genotypically-derived sex and documented phenotypic sex (n = 110), being genetic duplicates (n = 127), or mismatch among reported first-degree relative relationships (n = 470), comprising 707 total exclusions, approximately 3.5% of the original sample. PC analysis was performed using flashPCA software [42], and the first two PCs were used to classify individuals' ancestry as either White European or South Asian. We restricted our study sample to those of genetically defined European ancestry, which left 7,610 individuals for analysis: 5,709 genotyped on the CE array (2,405 offspring, 3,302 mothers) and 1901 on GSA array (614 offspring, 728 mothers; note fathers were also included in this group).

Variant filtering was performed prior to imputation where steps included the removal of SNPs with a call rate <95%, MAF < 0.01 and HWE P < $1 \times 10^{-6}$, as well as palindromic SNPs, duplicated SNPs, indels, and non-autosomal SNPs. After QC, the number of variants totalled n = 252,170 CE European, n = 471,297 GSA European. Imputation of samples was performed using the Michigan imputation server (Minimac4) [33,34]. Phasing was performed using Eagle v2.4 [35]. Genotype imputation was performed using the HRC v1.1 as the reference panel [32]. After imputation, variants with poor estimated imputation accuracy ($R^2 < 0.3$) were removed.

## Imputation of *KIR* types

Imputation of *KIR* types was performed using the KIR*IMP software [24]. KIR*IMP provides estimates of gene copy number, *KIR A* or *B* haplotype, as well as gene-content haplotypes for each individual using a reference panel of 301 SNPs constructed from a U.K. cohort of European ancestry. Using the SNP intersection between the input dataset and the reference panel, i.e., only the SNPs shared between the input dataset and the reference panel, the imputation software fits a random-forest model [49] to the reference panel using only the intersecting SNPs. The software outputs the imputed *KIR* types with the most likely allele and its associated posterior probability at each locus along with estimates of the average imputation accuracy for each *KIR* locus. The accuracy estimates are the out-of-bag (OOB) accuracy calculated during the

model-fitting process, which are shown to mimic cross-validation error rates and thus provide reliable estimates of KIR imputation performance – see Vukcevic et al. 2015 [24].

For each cohort, SNP genotypes from a segment of chromosome 19 encompassing the *KIR* region (53,000,000–58,000,000 bp in human genome build 37) were first extracted using the plink2 [50,51] software. To prioritise *KIR* imputation accuracy, we used the tools for the initial SNP genotype imputation (software and reference panel) that provided the greatest level of accuracy in each individual study. Although these tools varied among the studies, it ensured maximum SNP genotype estimation accuracy, which had important consequences for the success of the *KIR* haplotype estimation. SNP variants used in the KIR*IMP reference panel were then extracted from the imputed genotype cohort data and checked for consistency of allele frequency. Variants with discrepancy in frequency between the imputed SNP and the KIR reference panel were excluded prior to *KIR* imputation. These included SNPs with a difference in allelic frequency between the cohort and KIR reference panel greater than 10% or those with ambiguity around the 50% frequency mark, i.e., where both alleles within the cohort dataset had a frequency ≥45% but ≤55% or, when comparing to the reference panel, the same allele had a frequency ≤55% in one dataset and ≥45% in the other. Any palindromic SNPs exhibiting errors in strand alignment were flipped to align with the reference panel. Complete.haps and.sample files for the UKB, EFSOCH, and HAPO cohorts were then uploaded to the KIR*IMP server. Due to data sharing permissions for the ALSPAC and BiB cohorts, *KIR* imputation for these datasets necessitated the setup of a local KIR*IMP server environment where the imputation was performed using the software within the University of Exeter's High Performance Computing cluster rather than uploading the files to the external KIR*IMP server. From the imputation output, we used the posterior probabilities assigned to each allele to calculate a dosage which was then used to model each individual's estimated *KIR* type.

## Imputation of *HLA* types

In the UKB, *HLA* alleles were imputed to a multi-population reference panel centrally by UKB using the HLA*IMP:02 [25] software. *HLA* genotypes were imputed at four-digit (also known as two-field) resolution for 11 classical *HLA* genes, indicating both the serological antigen grouping or allele family (first field, e.g., *HLA-C*01*) and the subtype of the allele encoding the specific HLA protein (second field, e.g., *HLA-C*01:01*) [52]. A quality metric (Q) is reported of the absolute posterior probability of the allele inference for each genotype call [29]. We then used Q to determine a hard call for each HLA genotype to use in our association analyses. For EFSOCH, ALSPAC, and BiB, *HLA* alleles were imputed locally using SNP2HLA [26] software from which the dosages for the *HLA-C* alleles were extracted for analysis. As the BiB data were originally genotyped on the two different arrays, imputation was performed for each group separately and then merged into one dataset prior to association analyses. We imputed *HLA* data for the HAPO study, using a multi-ethnic HLA reference panel and the Michigan imputation server (Minimac4) [27], applied to extracted genotype data from 26,000,000–34,000,000 bp (human genome build 37) on chromosome six. After *HLA* data for each cohort was imputed and the dosage data or Q-values were extracted specific to the necessary *HLA-C* alleles, all dosages/Q-values were converted to hard calls using set thresholds as follows: 0 = dosage ≤ 0.3; 1 = dosage ≥ 0.7 and ≤1.3; 2 = dosage ≥1.7, otherwise set to missing. For association analyses, we derived *HLA-C1* or *-C2* group alleles according to their allele families based on the amino acid at position 80, with the C1 ligand carrying asparagine (*HLA-C*01*, *03*, *07*, *08*, *12*, *14*, *16* alleles), and C2 carrying lysine (*HLA-C*02*, *04*, *05*, *06*, *15*, *17*, *18* alleles) in alignment with previously published studies [10,11].

A summary of all genotype and *HLA* imputation software used, genotype imputation reference panels used, amount of *KIR* region SNP overlap with the KIR*IMP reference panel, final number of SNPs used for KIR imputation, and *KIR* imputation sample sizes are found in **Table 1**.

## Association between maternal *KIR* types, fetal *HLA-C*, and offspring birth weight

We performed multivariable mixed linear regression to explore the association between offspring birth weight and maternal *KIR A* vs *B* haplotypes (*AA, AB/BA, BB* genotypes) as well as the amount of *KIR2DS1* (0, 1, 2 copies of the gene)

**Table 1. Overview of genotyping, *HLA*, and *KIR* imputation materials.**

| Cohort | Genotype imputation software | Genotype imputation reference panel | *HLA* imputation software | No. SNPs to start (panel overlap) | No. SNPs for *KIR* imputation | N individuals |
|---|---|---|---|---|---|---|
| UKB | Michigan server | HRC v1.1 | HLA*IMP:02 | 271 | 259 | 8,498 |
| EFSOCH | TOPMed | TOPMed | SNP2HLA | 273 | 256 | 2,664* |
| HAPO | TOPMed | TOPMed | Minimac4 | 273 | 258 | 1,734 |
| ALSPAC | Michigan server | HRC v1.1 | SNP2HLA | 271 | 235 | 17,827 |
| BiB CE | Michigan server | HRC v1.1 | SNP2HLA | 254 | 229 | 5,709 |
| BiB GSA | Michigan server | HRC v1.1 | SNP2HLA | 261 | 230 | 1,901* |

*At the imputation stages, fathers were included in the EFSOCH and BiB GSA samples.

**Abbreviations:** UKB, UK Biobank; EFSOCH, Exeter Family Study of Childhood Health; HAPO, Hyperglycemia and Adverse Pregnancy Outcomes study; ALSPAC, Avon Longitudinal Study of Parents and Children; BiB, Born in Bradford; CE, Core Exome; GSA, Global Screening Array; HRC, Haplotype Reference Consortium; HLA, human leukocyte antigen; SNP, single nucleotide polymorphism; KIR, killer immunoglobulin-like receptor.

in the mothers under 3 different conditions of fetal *HLA*: in the presence of fetal *HLA-C2* (coded as 0 or 1 for *C2* group allele), in the presence of more *HLA-C2* alleles in the fetus relative to the mother (i.e. m*C1C1*/f*C1C2* and m*C1C2*/f*C2C2*), and when fetal *HLA-C2* is inferred to be inherited from the father (i.e. m*C1C1*/f*C1C2*); models are described in **Table 2**. These groupings of subjects and the combinations analyzed were defined in accordance with those previously investigated by Hiby et al. 2014 [19]. For inferred inheritance of fetal *HLA-C2* from the father, mother-child pairs where both mother and offspring were heterozygous (*mC1C2*/*fC1C2*) were not included in the analysis as paternal origin could not be determined unambiguously. Mixed linear regression models were used to account for mothers with multiple offspring in the sample, with mother's ID set as a random effect. Our study sample included only singleton pregnancies from live births carried to term (≥37 weeks and <43 weeks) according to gestational age data where available (information on gestational age is unavailable in UKB). For each cohort prior to modelling, we used its sample size to estimate the number of standard deviations within which most birth weight observations are expected to fall (EFSOCH = 3, HAPO = 4, ALSPAC = 4, BiB = 4 SDs, respectively). Birth weight observations outside of this range were classified as outliers and excluded prior to analysis (observations excluded: EFSOCH = 0, HAPO = 474, ALSPAC = 138, BiB = 5). For UKB specifically, where there were individuals with birth weight self-reported at multiple assessment centre visits, we took the mean of the multiple values, otherwise taking the single visit where they had data. If there was a difference in absolute value of multiple reported birth weights ≥1 kg, the individual was excluded. Birth weights <2.5 kg or >4.5 kg were also excluded from analysis for the UKB sample to reduce bias from possible reporting errors and to ensure, as far as possible, that preterm births were removed as information on gestational age was not available. This left final mother-offspring pair sample sizes for performing association analyses as follows: UKB = 3,165, EFSOCH = 625, HAPO = 470, ASLPAC = 4,550, and BiB = 1,792 (**Table 3**). All models were adjusted for offspring sex, gestational age (except UKB), genotyping batch, the first five genome-wide principal components (PCs) in the mother and offspring, and year of birth (UKB only). All analyses were conducted separately in each cohort before performing a fixed effect, inverse-variance weighted meta-analysis to estimate the interaction effects between maternal KIR and fetal HLA on birth weight across the entire study population. Characteristics of each population sample are described in **Table 3**.

## Results

### Quality of *KIR* imputation

The *KIR* types across the five study populations were well imputed, achieving both high estimated imputation accuracy and posterior probabilities overall. Imputation accuracy was estimated at over 95% for both the *KIR2DS1* locus and the *A* vs. *B* haplotype, with the exception of imputation of the *A* vs. *B* haplotype in the ALSPAC and BiB cohorts where accuracy

**Table 2. Models of the associations tested between maternal *KIR* types, fetal *HLA-C2*, and offspring birth weight.**

| Model | Genetic Effects Modelled | Formula |
|---|---|---|
| (1) | Presence of fetal *C2* Maternal *KIR B* Interaction | $BW_i \sim \beta_0 + \beta_1 * C2_{fi} + \beta_2 * KIRB_{mi} + \beta_3 * C2_{fi} * KIRB_{mi} + \beta_4 Cov_i + \varepsilon_i$ |
| (2) | Presence of fetal *C2* Maternal *KIR2DS1* Interaction | $BW_i \sim \beta_0 + \beta_1 * C2_{fi} + \beta_2 * KIR2DS1_{mi} + \beta_3 * C2_{fi} * KIR2DS1_{mi} + \beta_4 Cov_i + \varepsilon_i$ |
| (3) | Amount of fetal *C2* Maternal *KIR B* Interaction | $BW_i \sim \beta_0 + \beta_1 * PO_{fi} + \beta_2 * KIRB_{mi} + \beta_3 * PO_{fi} * KIRB_{mi} + \beta_4 Cov_i + \varepsilon_i$ |
| (4) | Amount of fetal *C2* Maternal *KIR2DS1* Interaction | $BW_i \sim \beta_0 + \beta_1 * PO_{fi} + \beta_2 * KIR2DS1_{mi} + \beta_3 * PO_{fi} * KIR2DS1_{mi} + \beta_4 Cov_i + \varepsilon_i$ |
| (5) | Paternal fetal *C2* Maternal *KIR B* Interaction | $BW_i \sim \beta_0 + \beta_1 * PatC2_{fi} + \beta_2 * KIRB_{mi} + \beta_3 * PatC2_{fi} * KIRB_{mi} + \beta_4 Cov_i + \varepsilon_i$ |
| (6) | Paternal fetal *C2* Maternal *KIR2DS1* Interaction | $BW_i \sim \beta_0 + \beta_1 * PatC2_{fi} + \beta_2 * KIR2DS1_{mi} + \beta_3 * PatC2_{fi} * KIR2DS1_{mi} + \beta_4 Cov_i + \varepsilon_i$ |

$BW_i$ = Offspring birth weight in grams.

$C2_{fi}$ = Fetal possession of at least one *HLA-C2* group allele; coded as 0 or 1 for absence or presence.

$KIRB_{mi}$ = Dosage of maternal *B* haplotype; calculated to correspond to *AA* = dosage of 0, *AB/BA* = 1, *BB* = 2 (zero uncertainty).

$KIR2DS1_{mi}$ = Dosage of maternal *KIR2DS1*; calculated to correspond to 0 copies = dosage of 0, 1 copy = 1, 2 copies = 2 (zero uncertainty).

$PO_{fi}$ = Parental origin; determining parental origin of fetal *C2* to test the effect of an extra copy of fetal *C2* compared to the mother (m*C1C1*/f*C1C2* and m*C1C2*/f*C2C2*); coded as less *C2* in fetus than in mother = 0, equal *C2* in mother and fetus = 1, and more *C2* in fetus than in mother = 2.

$PatC2_{fi}$ = Fetal *HLA-C2* is inferred to be inherited from the father (m*C1C1*/f*C1C2*); coded as 1 for paternally inherited *C2* allele, otherwise 0.

$Cov_i$ = included covariates.

$\varepsilon_i$ = residual errors.

was 81% accuracy (**Table 4**). The ALSPAC and BiB genotype data was more limited in its ability to be imputed due to the coverage of this region on the genotyping chip and thus did not achieve as high a broad haplotype estimation but still provided reasonable confidence at an accuracy over 80%.

**Little evidence of association between offspring birth weight and maternal *KIR* types or fetal *HLA-C2***

After meta-analysing the maternal *KIR* and fetal *HLA* effects on birth weight across the five cohorts of mother-offspring pairs, no interaction effect was observed between either the maternal *A* vs. *B* haplotype or the maternal *KIR2DS1* locus and fetal *HLA-C2* when *C2* was present in the fetus (yes vs. no; models 1 & 2), when the fetus had more *C2* compared to the mother (models 3 & 4), and when fetal *C2* was paternal in origin (models 5 & 6). Under each condition of fetal *HLA-C2*, no change in offspring birth weight was found for each additional maternal *B* allele or *KIR2DS1* allele, with nearly all effect estimates close to the null with wide confidence intervals (**Figs 1** and **2**, respectively). When performing a sensitivity analysis removing the UKB cohort, as the only sample with birth weight not adjusted for gestational age, the results were unchanged (see S1 and S2 Figs).

**Discussion**

In our investigation combining data from five different study populations, we found little evidence that maternal *KIR* haplotype was associated with a change in offspring birth weight in the presence of fetal *HLA-C2*. There were also no

**Table 3. Descriptive characteristics for participants in each cohort.**

| Characteristic | Mean (SD) for continuously measured variables and n (%) for categorical variables | | | | |
|---|---|---|---|---|---|
| | UKB | EFSOCH | HAPO | ALSPAC | BiB |
| Offspring birth weight (g) | 3,378 (411) | 3,524 (473) | 3,424 (394) | 3,495 (468) | 3,467 (487) |
| Offspring sex | | | | | |
| Male | 2,046 (64.5) | 327 (52.3) | 239 (50.9) | 2,231 (49.0) | 928 (51.8) |
| Female | 1,128 (35.5) | 298 (47.7) | 231 (49.1) | 2,319 (51.0) | 864 (48.2) |
| Gestational age at birth (weeks) | NA | 40.1 (1.22) | 40.0 (1.17) | 39.8 (1.26) | 39.6 (1.18) |
| Maternal age (years) | 23.1 (2.65) | 30.4 (5.25) | 30.1 (5.56) | 29.1 (4.59) | 27.2 (5.89) |
| Maternal BMI (kg/m$^2$)* | NA | 24.1 (4.32) | 28.3 (4.65) | 21.3 (7.20) | 27.0 (5.97) |
| Maternal height (cm) | 160.3 (5.95) | 165.0 (6.36) | 164.1 (6.43) | 161.5 (22.95) | 164.6 (6.15) |
| Maternal smoking status | | | | | |
| Never | 1833 (57.8) | 431 (69.0) | 408 (86.8) | 2,452 (53.9) | 714 (39.8) |
| Ever | 1316 (41.5) | 173 (27.7) | 62 (13.2) | 2,023 (44.5) | 947 (52.8) |
| Unknown | 25 (0.8) | 21 (3.3) | -- | 75 (1.6) | 131 (7.3) |
| Parity | NA | | | | |
| 1st child | | 272 (43.5) | 334 (71.1) | 2,049 (45.0) | 901 (50.3) |
| 2nd child or greater | | 353 (56.5) | 136 (28.9) | 2,453 (53.9) | 891 (49.7) |
| Unknown | | -- | -- | 48 (1.1) | |
| **Total n** | 3,165 | 625 | 470 | 4,550 | 1,792 |

*Pre-pregnancy BMI, except for the HAPO and BiB cohorts where BMI was obtained from measurements at 28 weeks gestation.

**Abbreviations:** UKB, UK Biobank; EFSOCH, Exeter Family Study of Childhood Health; HAPO, Hyperglycemia and Adverse Pregnancy Outcome study; ALSPAC, Avon Longitudinal Study of Parents and Children; BiB, Born in Bradford; SD, standard deviation; BMI, body mass index.

**Table 4. Estimated *KIR* imputation accuracy across all cohorts for the *KIR A* vs. *B* haplotype and *KIR2DS1* locus.**

| Locus | Estimated Accuracy (%) | | | | | |
|---|---|---|---|---|---|---|
| | UKB | EFSOCH | HAPO | ALSPAC | BiB CE | BiB GSA |
| *KIR A* vs. *B* | 98.75 | 96.03 | 96.03 | 80.79 | 81.0 | 82.88 |
| *KIR2DS1* | 98.33 | 98.54 | 98.75 | 96.66 | 96.45 | 96.87 |

**Abbreviations:** UKB, UK Biobank; EFSOCH, Exeter Family Study of Childhood Health; HAPO, Hyperglycemia and Adverse Pregnancy Outcomes study; ALSPAC, Avon Longitudinal Study of Parents and Children; BiB, Born in Bradford; CE, Core Exome; GSA, Global Screening Array.

associations observed with birth weight when examining the dosage of the activating receptor-encoding *KIR2DS1* gene, when the fetus possessed more *HLA-C2* than the mother, or when the fetal *HLA-C2* was paternal in origin. This contrasts with the findings of Hiby et al. 2014 [19] where the presence of maternal *KIR2DS1* combined with fetal *C2* of paternal origin resulted in an average increase in birth weight of ~250 g (total n = 1,316 pregnancies) compared to offspring without maternal *KIR2DS1* and less or equal fetal *C2*; we observed a null effect in a sample of approximately 10,600 pairs, which was an order of magnitude larger than that of Hiby et al.

We aimed to replicate these associations but were unable to find evidence of an association between *HLA* and *KIR* combinations with birth weight. Rather than test presence vs. absence of maternal *KIR2DS1*, we elected to code the variable as gene copy number as output by the *KIR* imputation software; however, we would not expect this to alter the ability to detect an association with birth weight. We additionally tested *HLA-C* in combination with a maternal *KIR A* vs *B* haplotype as this is representative of the inhibitory vs activating genotype. We would expect this to reflect a similar

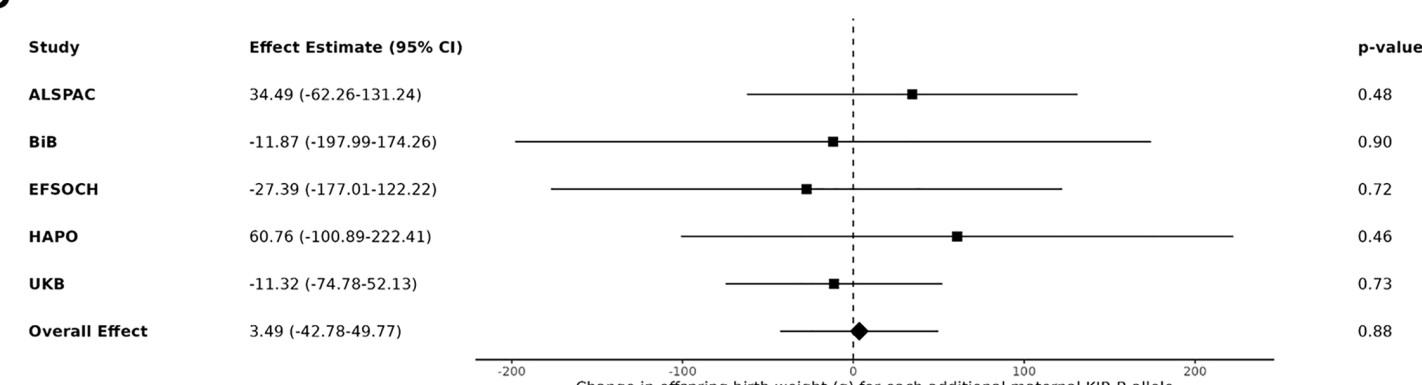

**Fig 1. Meta-analysis of the estimated change in offspring birth weight (g) for each additional maternal *KIR B* allele. (A)** In the presence of fetal *HLA-C2* (n = 10,591). **(B)** In the presence of more *HLA-C2* alleles in the fetus relative to the mother (n = 10,562). **(C)** When fetal *HLA-C2* is paternal in origin (n = 10,589).

pattern of association as with the *KIR2DS1* locus on its own, perhaps with some attenuation as *KIR2DS1* may not be present on all individuals but similarly found no association in this variation of analysis. Some key differences to note between these two studies include the fact that we imputed both *KIR* and *HLA* types in our study, whereas Hiby et al. [19] were able to directly type mothers and babies in their smaller sample. Our imputation estimated *KIR* and *HLA* alleles

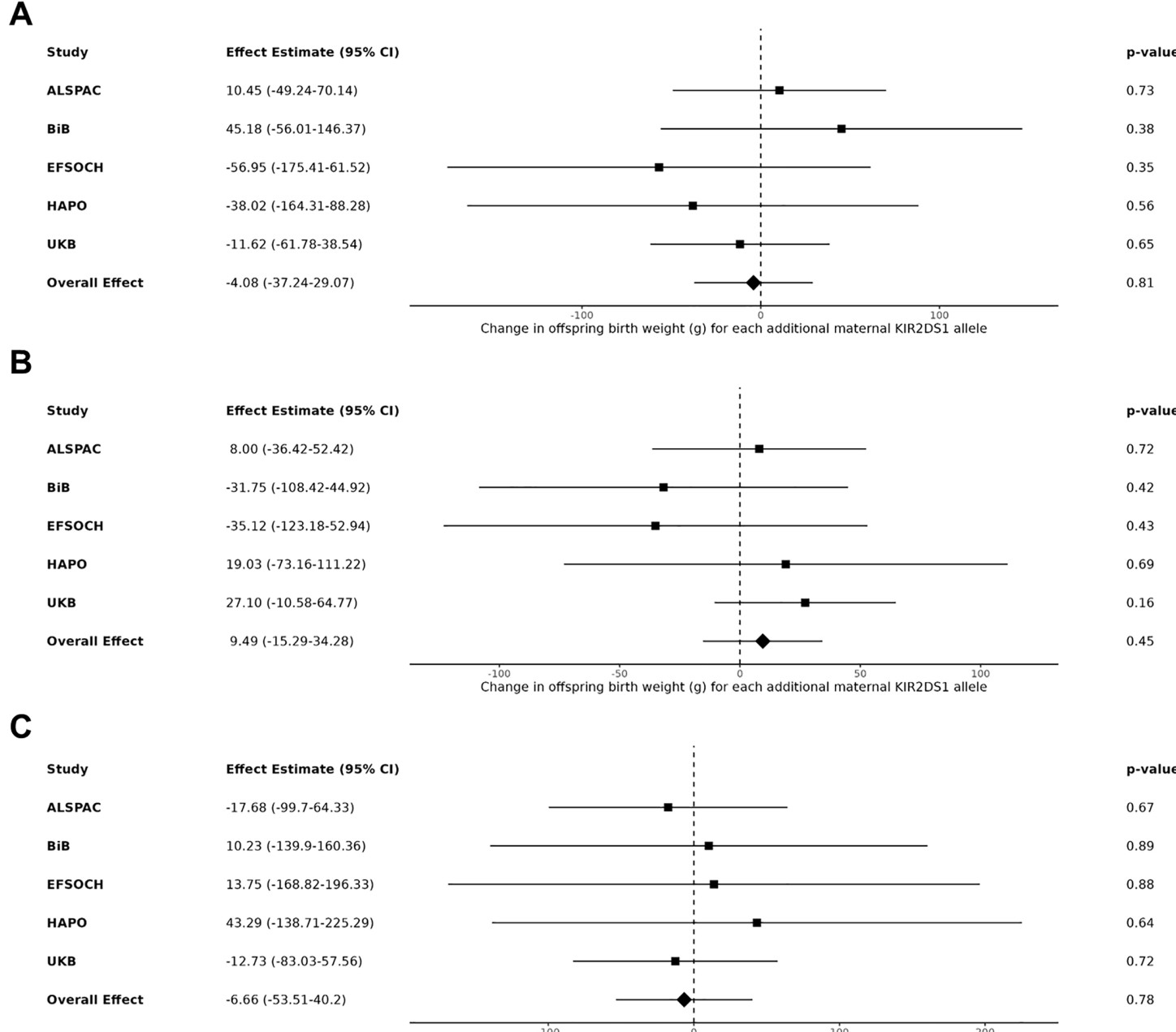

**Fig 2. Meta-analysis of the estimated change in offspring birth weight (g) for each additional maternal *KIR2DS1* allele. (A)** In the presence of fetal *HLA-C2* (n = 10,591). **(B)** In the presence of more *HLA-C2* alleles in the fetus relative to the mother (n = 10,562). **(C)** When fetal *HLA-C2* is paternal in origin (n = 10,589).

with high accuracy, but there will be some level of uncertainty in the genotype calls made for individuals; any genotyping errors will bias results towards the null. We also removed birth weight outliers on both ends of the spectrum, keeping our sample to uncomplicated pregnancies without extreme birth weight babies. Hiby et al. [19], though also investigating uncomplicated pregnancies, investigated birth weights >5th centile with opportunity to include high birth weights if

clinically determined to have no known medical cause, therefore including a greater range of birth weights than our study. Even so, in our included measures of birth weight covering normal pregnancy, we should expect to see the same previously observed patterns of association if the underlying biological mechanisms are acting as proposed. Given the null findings in our much larger sample, exclusion of outliers in our analyses alone is not sufficient to explain the lack of an association if a true effect of 250g higher birth weight in maternal *KIR2DS1* carriers and fetal carriers of paternal *HLA-C2* exists.

Previous work has been done to determine modelling methods that most accurately capture maternal-fetal genotype (MFG) interactions, such as that by Clark et al. 2016 and 2017, which developed the extended quantitative-MFG (QMFG) test, a linear mixed model that can apply familial multi-locus genotype data [53,54]. Their model was designed to address the issues of low power or incorrect conclusions obtained when using standard models that only consider offspring genotypes [53]. In the Clark et al. 2017 study, they used this model to the investigate the same relationship between maternal *KIR2DS1*, more fetal *HLA-C2* than the mother, and birth weight as in Hiby et al. 2014, using the same sample of white European individuals (n ≈ 1,300 mother-child pairs) [54]. They expanded this replication by specifically testing an interaction between the loci and if there are additional independent effects of each locus. They found that the model that best fit this data and assessed these relationships, models the genetic effects of maternal *KIR2DS1*, more *HLA-C2* in the fetus, and an interaction between them (denoted as Model 1 in their study). With this model, they found a significant *KIR–HLA-C* interaction effect on birth weight that explained ∼1% of the phenotypic variation in birth weight [54]. The model they used is essentially Model 4 in our study presented in **Table 2** and **Fig 2b**: a linear mixed model testing the same genetic effects with both loci included as well as an interaction term. With respect to covariate adjustment, we adjusted our tested models for offspring sex and gestational age (except UKB) as they did, with the added benefit of adjustment for mother and child principal components in our models to guard against effects of population stratification. We have thus used a method previously determined to be the most appropriate to assess the maternal-fetal genetic relationships at these loci but were unable to replicate an effect in our analysis. It is important to reiterate that although Clark et al. 2017 were able to replicate an association between maternal *KIR2DS1*, greater *HLA-C2* in the fetus, and increased birth weight as in the Hiby et al. 2014 study, they used the same small sample of mother-child pairs while our study increased sample size by approximately a factor of 10. Therefore, regardless of model similarities or differences, the impact of sample size on results must be considered.

The most likely explanation for our lack of evidence of a relationship between maternal *KIR* and fetal *HLA-C* on birth weight is that the previous association was a false-positive result, driven partly by the relatively small sample. It is well established that the lower a study's power, the more likely a positive finding is to be false [55]. To our knowledge, Hiby et al. [19] also did not control for population substructure as we have done here with the inclusion of ancestry PCs. Unaccounted for differences in ancestry, even within a sample restricted to a single continent of origin, is a common reason for false-positive findings due to allele frequency differences between population subgroups among cases and controls [56]. The effect sizes observed previously [19] are not realistic for common complex traits and inconsistent with the effect sizes observed for common variants for birth weight and other traits that have been investigated over the years with GWAS [21,57,58]. Though our confidence intervals are wide, they are small enough to rule out the effect size found by Hiby et al. [19].

It is possible that the published associations of *KIR* and *HLA-C2* are true with respect to pregnancy complications and the extremes of birth weight. Those pregnancies would be much more likely to be influenced by placental insufficiency or dysfunction which could be what these genes and their molecules truly influence rather than weight of the fetus per se. Though strides have been made with respect to GWAS and identification of genetic variants influencing pregnancy belonging to the mother, fetus, or shared by both [20,21,59–61], there is still much work to be done in the exploration of maternal-fetal interaction and how genes work together to influence or cause birth and healthy aging outcomes. New and emerging large scale genetic studies identifying variants associated with phenotypes such as pre-eclampsia [62] and

placental weight [63], may offer novel avenues of investigation in terms of testable instruments and sizeable population samples to draw from. It is also worth noting that a recent study able to partition maternal and fetal effects of classical HLA alleles on birth weight found only a nominal association (P<0.05) of *HLA-C* on birth weight while certain *HLA-A* and *HLA-B* alleles showed a stronger association (P<0.001) with birth weight after Bonferroni correction for multiple testing [61]. Evaluation of immunogenetic relationships would additionally benefit from the use of methods which could type individuals at these challenging loci even more accurately on a larger scale. The rise of high throughput whole genome sequencing technologies makes the availability of such data likely in coming years.

We are limited in our ability to generalize these findings to non-European populations as we did not use an ancestrally diverse set of individuals. Some additional limitations exist within the *KIR* imputation software itself and how it applies the data to be imputed. The estimated accuracy of the *KIR* loci is calculated based on the level of intersection between the reference panel and the input data [24]. It is highly dependent on how many of the 301 reference SNPs overlap with those in the study population sample, with 12 of these SNPs deemed "highly informative". The accuracy figures in **Table 4** are an estimate of how well each *KIR* type can expect to be imputed using SNP genotype data coming from the same population as the reference panel. The accuracy in actuality may be worse if the underlying ancestry of the population sample differs from that of the reference panel or if the genotype quality is poor. This is reflected in how well the distributions of the posterior probabilities of the most likely allele at each *KIR* locus align between the reference panel and the input dataset, for which our results largely showed very good concordance – an indicator that the KIR loci are imputed well, though less so for *KIR A* vs. *B* in the ALSPAC and BiB cohorts likely due to poor SNP coverage of the genotyping chip in this region (see S3–S8 Figs). Adding further support for our study findings relative to our imputation quality, we performed a sensitivity analysis using simulations to examine the effect of varying imputation accuracy on the power to detect a main effect of maternal *KIR2DS1* as well as an interaction between maternal *KIR2DS1* and fetal *HLA-C2* on offspring birth weight. The results indicated that we maintain very high power to detect main effects and interactions in our study, even with substantially reduced KIR imputation accuracy (simulated as low as 70%; see S1 Text). This can be attributed to having a sample size an order of magnitude larger than the original Hiby et al. study. Lastly, we sought to replicate initial published findings by Hiby et al. [19] tested in a population of uncomplicated pregnancies; we did not test these same associations in complicated pregnancies with a greater spectrum of extremes, e.g., with very low or high birth weights and pre-eclampsia, where associations could be more evident if they reflect the known underlying biological mechanisms.

Overall, though observational studies provide evidence for the role of maternal KIR and fetal HLA-C2 in pregnancy outcomes impacted by placental dysfunction, and by extension birth weight, we were unable to detect evidence of an association of these receptor-ligand pairs with offspring birth weight. Our findings here reinforce the importance of replication and use of the largest sample sizes available to maximize power, as not all observed genetic associations may be valid, even with strong biological plausibility. More research is needed to validate this relationship, or lack thereof, in more diverse populations and in complicated pregnancy cases in similarly robust sample sizes. Having successfully demonstrated the use of widely available SNP genotype data from large population studies to estimate individuals' *KIR* and *HLA* genotypes, new statistical methods along with growing biobanks and consortia should be harnessed to better inform robust conclusions regarding maternal-fetal genetic interplay.

## Supporting information

**S1 Checklist. STROBE checklist for reporting of cross-sectional studies.** von Elm E, Altman DG, Egger M, Pocock SJ, Gøtzsche PC, Vandenbroucke JP; STROBE Initiative. The Strengthening the Reporting of Observational Studies in Epidemiology (STROBE)statement: guidelines for reporting observational studies. PLoS Med. 2007 Oct 16;4(10):e296. PMID: 17941714.
(PDF)

**S1 Text. Simulation: Estimating power to detect a main effect of *KIR2DS1* and an interaction between maternal *KIR2DS1* and fetal *HLA-C2* on offspring birth weight with varying imputation accuracy.**
(PDF)

**S1 Fig. Meta-analysis omitting the UKB cohort (the only sample with birth weight not corrected for gestational age) to assess impact on the association between maternal *KIR B alleles*, fetal *HLA-C,* and birth weight.** Effects represent the estimated change in offspring birth weight (g) for each additional maternal *KIR B* allele **(A)** in the presence of fetal *HLA-C2*, **(B)** in the presence of more *HLA-C2* alleles in the fetus relative to the mother, and **(C)** when fetal *HLA-C2* is paternal in origin; total n = 7,437. UKB, UK Biobank; EFSOCH, Exeter Family Study of Childhood Health; HAPO, Hyperglycemia and Adverse Pregnancy Outcomes study; ALSPAC, Avon Longitudinal Study of Parents and Children; BiB, Born in Bradford.
(PDF)

**S2 Fig. Meta-analysis omitting the UK Biobank (UKB) cohort (the only sample with birth weight not corrected for gestational age) to assess impact on the association between maternal *KIR2DS1 alleles*, fetal *HLA-C,* and birth weight.** Effects represent the estimated change in offspring birth weight (g) for each additional maternal *KIR2DS1* allele **(A)** in the presence of fetal *HLA-C2*, **(B)** in the presence of more *HLA-C2* alleles in the fetus relative to the mother, and **(C)** when fetal *HLA-C2* is paternal in origin; total n = 7,437. UKB, UK Biobank; EFSOCH, Exeter Family Study of Childhood Health; HAPO, Hyperglycemia and Adverse Pregnancy Outcomes study; ALSPAC, Avon Longitudinal Study of Parents and Children; BiB, Born in Bradford.
(PDF)

**S3 Fig. KIR imputation plots for UK Biobank (UKB) mother-offspring pairs (n = 8,498 at imputation stage). (A)** The estimated imputation accuracy achieved at each KIR locus**. (B)** The distribution of the posterior probabilities of the most likely alleles for each KIR locus where the distribution of the input dataset is shown in blue against the reference panel distribution shown in red.
(PDF)

**S4 Fig. KIR imputation plots for the Exeter Family Study of Childhood Health (EFSOCH) mother-offspring pairs (n = 2,664 at imputation stage). (A)** The estimated imputation accuracy achieved at each KIR locus. **(B)** The distribution of the posterior probabilities of the most likely alleles for each KIR locus where the distribution of the input dataset is shown in blue against the reference panel distribution shown in red.
(PDF)

**S5 Fig. KIR imputation plots for the Hyperglycemia and Adverse Pregnancy Outcome (HAPO) study mother-offspring pairs (n = 1,734 at imputation stage). (A)** The estimated imputation accuracy achieved at each KIR locus. **(B)** The distribution of the posterior probabilities of the most likely alleles for each KIR locus where the distribution of the input dataset is shown in blue against the reference panel distribution shown in red.
(PDF)

**S6 Fig. KIR imputation plots for the Avon Longitudinal Study of Parents and Children (ALSPAC) mother-offspring pairs (n = 17,827 at imputation stage). (A)** The estimated imputation accuracy achieved at each KIR locus. **(B)** The distribution of the posterior probabilities of the most likely alleles for each KIR locus where the distribution of the input dataset is shown in blue against the reference panel distribution shown in red.
(PDF)

**S7 Fig. KIR imputation plots for the Born in Bradford (BiB) Core Exome mother-offspring pairs (n = 5,709 at imputation stage). (A)** The estimated imputation accuracy achieved at each KIR locus**. (B)** The distribution of the posterior

probabilities of the most likely alleles for each KIR locus where the distribution of the input dataset is shown in blue against the reference panel distribution shown in red.
(PDF)

**S8 Fig. KIR imputation plots for the Born in Bradford (BiB) Global Screening Array mother-offspring pairs (n = 1,901 at imputation stage). (A)** The estimated imputation accuracy achieved at each KIR locus. **(B)** The distribution of the posterior probabilities of the most likely alleles for each KIR locus where the distribution of the input dataset is shown in blue against the reference panel distribution shown in red.
(PDF)

## Acknowledgments

This research has been conducted using the UK Biobank Resource under application number 53641.

This study represents independent research supported by the National Institute of Health Research (NIHR) Exeter Clinical Research Facility. The views expressed are those of the author(s) and not necessarily those of the NHS, the NIHR or the Department of Health and Social care.

The Exeter Family Study of Childhood Health (EFSOCH) was supported by South West NHS Research and Development, Exeter NHS Research and Development, the Darlington Trust and the Peninsula NIHR Clinical Research Facility at the University of Exeter. The opinions given in this paper do not necessarily represent those of NIHR, the NHS or the Department of Health. We would like to acknowledge Andrew Hattersley as the principal investigator, and Bea Knight for her contribution to data collection, of the EFSOCH study.

We are extremely grateful to all the families who took part in this study, the midwives for their help in recruiting them, and the whole ALSPAC team, which includes interviewers, computer and laboratory technicians, clerical workers, research scientists, volunteers, managers, receptionists and nurses. This publication is the work of the authors and C.S.D and R.M.F will serve as guarantors for the contents of this paper.

Born in Bradford is only possible because of the enthusiasm and commitment of the children and parents in BiB. We are grateful to all the participants, health professionals, schools and researchers who have made Born in Bradford happen.

This project utilised high-performance computing funded by the UK Medical Research Council (MRC) Clinical Research Infrastructure Initiative (award number MR/M008924/1).

## Author contributions

**Conceptualization:** Caitlin Stephanie Decina, Nicole M. Warrington, Robin N. Beaumont, Rachel M. Freathy, David M. Evans.

**Data curation:** Caitlin Stephanie Decina, William L. Lowe Jr.

**Formal analysis:** Caitlin Stephanie Decina, Nicole M. Warrington, Robin N. Beaumont, David Squire.

**Funding acquisition:** Rachel M. Freathy, David M. Evans.

**Methodology:** Caitlin Stephanie Decina, Nicole M. Warrington, Robin N. Beaumont, Beilei Bian, Caroline Brito Nunes, Geng Wang, David Squire, Damjan Vukcevic, Stephen Leslie, Rachel M. Freathy, David M. Evans.

**Project administration:** Rachel M. Freathy, David M. Evans.

**Resources:** David Squire.

**Software:** David Squire, Damjan Vukcevic, Stephen Leslie.

**Supervision:** Nicole M. Warrington, Robin N. Beaumont, Rachel M. Freathy, David M. Evans.

**Validation:** Caitlin Stephanie Decina.

**Visualization:** Caitlin Stephanie Decina.

**Writing – original draft:** Caitlin Stephanie Decina.

**Writing – review & editing:** Caitlin Stephanie Decina, Nicole M. Warrington, Robin N. Beaumont, Beilei Bian, Caroline Brito Nunes, Geng Wang, William L. Lowe Jr., David Squire, Damjan Vukcevic, Stephen Leslie, Rachel M. Freathy, David M. Evans.

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
