## [Decision Letter · Decision Letter 0]

28 Jul 2025

PGENETICS-D-25-00434

Examining the association between fetal *HLA-C*, maternal , maternal , maternal , maternal *KIR* haplotypes and birth weighthaplotypes and birth weighthaplotypes and birth weighthaplotypes and birth weight

PLOS Genetics

Dear Dr. Decina,

Thank you for submitting your manuscript to PLOS Genetics. After careful consideration, we feel that it has merit but does not fully meet PLOS Genetics's publication criteria as it currently stands. Therefore, we invite you to submit a revised version of the manuscript that addresses the points raised during the review process.

Please submit your revised manuscript within 60 days Aug 27 2025 11:59PM. If you will need more time than this to complete your revisions, please reply to this message or contact the journal office at plosgenetics@plos.org. Please include the following items when submitting your revised manuscript:

We look forward to receiving your revised manuscript.

Kind regards,

Kai Wang

Academic Editor

PLOS Genetics

Gregory Cooper

Section Editor

PLOS Genetics

Aimée Dudley

Editor-in-Chief

PLOS Genetics

Anne Goriely

Editor-in-Chief

PLOS Genetics

**Additional Editor Comments:**

Please clarify the imputation procedure, removal of extreme values, and the inclusion of covariates.

**Journal Requirements:**

At this stage, the following Authors/Authors require contributions: Caitlin S. Decina, Nicole M. Warrington, Robin N. Beaumont, Beilei Bian, Caroline Brito Nunes, Geng Wang, William L. Lowe, Jr., David Squire, Damjan Vukcevic, Stephen Leslie, Rachel M. Freathy, and David M. Evans. Please ensure that the full contributions of each author are acknowledged in the "Add/Edit/Remove Authors" section of our submission form.

The list of CRediT author contributions may be found here: https://journals.plos.org/plosgenetics/s/authorship#loc-author-contributions

5) Please ensure that the funders and grant numbers match between the Financial Disclosure field and the Funding Information tab in your submission form. Note that the funders must be provided in the same order in both places as well. State what role the funders took in the study. If the funders had no role in your study, please state: "The funders had no role in study design, data collection and analysis, decision to publish, or preparation of the manuscript.".

**Reviewers' comments:**

Reviewer's Responses to Questions

**Comments to the Authors:**

Reviewer #1: Decina. PLoS Genetics

KIR, HLA-C and birth weight

This manuscript has used several cohorts to look at the association of maternal KIR and fetal HLA-C variants with birthweight. The previous studies looking at this question have only had few samples. They have now used birthweight data from ~10,000 mother-offspring pairs from 5 different cohorts. The KIR and HLA polymorphisms have been imputed.

There are several issues that need addressing:

1. The cohorts are all slightly different. The gestational age is known, and the birth weight has been corrected for this in 4 cohorts, but not in UKB where the birthweight is also self reported. Gestational age is clearly strongly correlated with birth weight. This means this cohort should be left out in their analysis to see if the results are altered.

2. How were these pairs corrected for other known factors that influence birth weight – for example: BMI, fetal sex, first pregnancy. It would be helpful to have a flow chart of how all the exclusions were made.

3. It seems that birth weights at the extremes of the normal distribution have been excluded (<2.5Kg and >4.5Kg). As birth weight is under stabilizing selection this presents a significant problem as the effects would be expected at these extremes.

Related to this is that the previous studies on KIR and HLA-C in pregnancy have mainly looked at pregnancies presenting with clinical problems. These have mainly been preeclampsia and intra-uterine growth restriction – those at one end of the birth weight range.

Thus, the question posed in this study is rather different than the question addressed in the study they are supposedly disproving. It is perfectly possible that KIR and HLA-C variants have little influence in the middle of the birth weight spectrum. They are studying birthweights that are not at the extremes - but all previous studies have focussed on the extremes.

To reiterate, this study uses continuous variable EXCLUDING extremes with the assumption that the effect is linear, which is very likely not to be the case.

4. The work of Sinsheimer PMID: 26567478 PMID: 28214848 is not cited and must be considered as GWAS for quantitative traits can miss genetic associations. These models need to be included as they might show model specification and power. There might be too many variables in their models to properly test for interaction - mixing cohorts, parity, gestational age.

5. Key to this study is the typing at these extraordinarily polymorphic loci. KIR*IMP is still not considered robust by many labs. Only one method has so far been published to impute KIR reflecting the difficulties in doing this. It is stated that imputation accuracy is >95% but how is this determined? To confirm the findings, it is essential to type at least 25% of samples by routine KIR typing to confirm their accuracy of imputation. What is the percentage of correct calls when this has been done? Do their imputed KIR reflect published findings in the populations tested in the cohorts?

6. Only KIR2DS1 has been imputed but what about other activating KIR, especially KIR2DS4

7. Several groups have shown how uterine NK cells can alter invasive trophoblast behaviour that is influenced by KIR activation but this work is not cited.

Reviewer #2: The authors report an attempt to replicate the previously described effect of the maternal KIR haplotype on offspring birth weight, particularly in cases where the offspring inherited the paternal HLA-C2 haplotype. They have substantially increased the sample size by meta-analysing five independent cohorts of mother–offspring pairs, improving upon the original study by Hiby et al. (2014). While the introduction and methods are well structured, the results are somewhat disappointing, as no significant associations were observed. Nevertheless, it is valuable to publish well-conducted negative findings, particularly when they challenge previously reported associations, and such efforts should be more strongly encouraged in the field.

Overall, the authors did a good job of describing their experiments and discussing why and how their results differed from previously reported associations.

My main concern is the heterogeneity and potential limitations in how KIR and HLA imputation was performed across cohorts. The entire study relies on imputed KIR and HLA data, which the authors report as accurate (Table 4). However, it is unclear how imputation accuracy was evaluated. From my understanding, the best way to assess accuracy is to downsample WGS data to SNP array sites, impute back, and compare to the WGS calls. I assume WGS data were only available for UKBB, if at all — so what benchmark was used to report accuracy?

In the Study population section, it is described that each independent study was quality-controlled and imputed. For some studies (e.g., BiB), a post-imputation quality filter is specified (variants with imputation info score < 0.3 were removed). Was this also applied to the other cohorts?

It is also not entirely clear what was used as input for the KIR and HLA imputation. In the KIR*IMP manuscript, phased SNP array data are used as input. From reading your methods, it seems that you instead used imputed data as input. If so, I have a few follow-up questions:

• Why not use only genotyped variants as input? Was there insufficient overlap with the reference panel?

• Was the imputed data used as input already phased? Some imputation tools, such as IMPUTE5, output phased posterior probabilities — is this also the case when using the Michigan Imputation Server? From my understanding, KIR*IMP requires phased input.

• An imputation quality filter of 0.3 seems rather permissive. Many GWAS use a threshold of 0.8. In your case, since the imputed data were used as input for KIR imputation, I believe it would be even more important to apply stringent quality control, for example using only variants with info score > 0.99, which should still retain enough markers for KIR imputation.

Minor comments:

• Line 288: How was the parental origin of the fetal HLA-C2 determined, particularly when both mother and offspring were heterozygous C1/C2?

• Since mothers are genotyped, could you also explore the effect of the transmitted versus untransmitted maternal haplotypes?

In summary, I think this is a well-written and valuable study that addresses an important question with a much larger sample size than the original work. Although the lack of association is disappointing, this careful attempt to replicate the previous findings clearly shows how important replication is in genetic association studies, especially since the original study did not include any independent replication. Publishing well-conducted negative results, like in this case, is crucial and should be encouraged more in the field.

Reviewer #3: The manuscript by Decina et al examines the previous reported association between fetal human leukocyte antigen-C (HLA-C) and maternal killer immunoglobulin-like receptor (KIR) haplotypes on birth weight. This is important, as lower and higher birth weights have implications on health outcomes for both the mother and newborn. Previous associations with lower sample sizes suggested a strong association. The current study reexamines this association with a much larger cohort and finds no association. This is an important finding that will be of broad interest to human geneticists and physicians alike. The manuscript is well written, and the methods used are nicely described. The authors acknowledge that the cohort is skewed toward individuals of European descent and further studies are needed to thoroughly examine other cohorts.

**Have all data underlying the figures and results presented in the manuscript been provided?**

Large-scale datasets should be made available via a public repository as described in the *PLOS Genetics*
data availability policy, and numerical data that underlies graphs or summary statistics should be provided in spreadsheet form as supporting information., and numerical data that underlies graphs or summary statistics should be provided in spreadsheet form as supporting information., and numerical data that underlies graphs or summary statistics should be provided in spreadsheet form as supporting information., and numerical data that underlies graphs or summary statistics should be provided in spreadsheet form as supporting information.

Reviewer #1: Yes

Reviewer #2: Yes

Reviewer #3: Yes

PLOS authors have the option to publish the peer review history of their article (what does this mean?). If published, this will include your full peer review and any attached files.). If published, this will include your full peer review and any attached files.). If published, this will include your full peer review and any attached files.). If published, this will include your full peer review and any attached files.

...

Reviewer #1: No

Reviewer #2: No

Reviewer #3: No

**Figure resubmission:**
---

## [Decision Letter · Decision Letter 1]

20 Jan 2026

PGENETICS-D-25-00434R1

Examining the association between fetal *HLA-C*, maternal , maternal , maternal , maternal *KIR* haplotypes and birth weighthaplotypes and birth weighthaplotypes and birth weighthaplotypes and birth weight

PLOS Genetics

Dear Dr. Decina,

Thank you for submitting your manuscript to PLOS Genetics. After careful consideration, we feel that it has merit but does not fully meet PLOS Genetics's publication criteria as it currently stands. Therefore, we invite you to submit a revised version of the manuscript that addresses the points raised during the review process.

We look forward to receiving your revised manuscript.

Kind regards,

Kai Wang

Academic Editor

PLOS Genetics

Gregory Cooper

Section Editor

PLOS Genetics

Aimée Dudley

Editor-in-Chief

PLOS Genetics

Anne Goriely

Editor-in-Chief

PLOS Genetics

**Additional Editor Comments:**

The authors are expected to address the issues raised by a reviewer: KIRIMP is not readily available and its accuracy needs to be evaluated (by typing 25% of the sample); There is a lack of data on gestational age; ; Extremes of the normal birth weight were left out (so your work isn't a true replication study of Hiby et al.).

**Journal Requirements:**

**Reviewers' comments:**

Reviewer's Responses to Questions

**Comments to the Authors:**

Reviewer #1: There are still concerns about this paper.

The method for imputing KIR is not considered gold standard in the field.

Indeed, two of the three reviewers picked up on the issues with KIRIMP.

This method, KIRIMP, is not readily available, so it has been difficult for other labs to independently evaluate its accuracy. Many SNPs seem missing which will inevitably result in inaccuracies.

It is essential therefore that they type 25% of their samples to test its accuracy.

Only once this has been done will the data be convincing to a wide audience.

There are also still concerns about the lack of data on gestational age and leaving out the extremes of the normal birth weight distribution that have not been resolved.

Reviewer #2: I thank the authors for carefully considering all my comments. I believe the points I raised have been appropriately addressed.

**Have all data underlying the figures and results presented in the manuscript been provided?**

Large-scale datasets should be made available via a public repository as described in the *PLOS Genetics*
data availability policy, and numerical data that underlies graphs or summary statistics should be provided in spreadsheet form as supporting information., and numerical data that underlies graphs or summary statistics should be provided in spreadsheet form as supporting information., and numerical data that underlies graphs or summary statistics should be provided in spreadsheet form as supporting information., and numerical data that underlies graphs or summary statistics should be provided in spreadsheet form as supporting information.

Reviewer #1: Yes

Reviewer #2: Yes

PLOS authors have the option to publish the peer review history of their article (what does this mean?). If published, this will include your full peer review and any attached files.). If published, this will include your full peer review and any attached files.). If published, this will include your full peer review and any attached files.). If published, this will include your full peer review and any attached files.

...

Reviewer #1: No

Reviewer #2: No

**Figure resubmission:**
---

## [Editor Report · Decision Letter 2]

24 Mar 2026

Dear Dr Decina,

We are pleased to inform you that your manuscript entitled "Examining the association between fetal *HLA-C*, maternal , maternal , maternal , maternal *KIR* haplotypes and birth weight" has been editorially accepted for publication in PLOS Genetics. Congratulations!haplotypes and birth weight" has been editorially accepted for publication in PLOS Genetics. Congratulations!haplotypes and birth weight" has been editorially accepted for publication in PLOS Genetics. Congratulations!haplotypes and birth weight" has been editorially accepted for publication in PLOS Genetics. Congratulations!

Yours sincerely,

Gregory M. Cooper, PhD

Section Editor

PLOS Genetics

Gregory Cooper

Section Editor

PLOS Genetics

Aimée Dudley

Editor-in-Chief

PLOS Genetics

Anne Goriely

Editor-in-Chief

PLOS Genetics

BlueSky: @plos.bsky.social

Comments from the reviewers (if applicable):

**Data Deposition**

If you have submitted a Research Article or Front Matter that has associated data that are not suitable for deposition in a subject-specific public repository (such as GenBank or ArrayExpress), one way to make that data available is to deposit it in the Dryad Digital Repository. As you may recall, we ask all authors to agree to make data available; this is one way to achieve that. A full list of recommended repositories can be found on our . As you may recall, we ask all authors to agree to make data available; this is one way to achieve that. A full list of recommended repositories can be found on our . As you may recall, we ask all authors to agree to make data available; this is one way to achieve that. A full list of recommended repositories can be found on our . As you may recall, we ask all authors to agree to make data available; this is one way to achieve that. A full list of recommended repositories can be found on our website....

http://datadryad.org/submit?journalID=pgenetics&manu=PGENETICS-D-25-00434R2

Additionally, please be aware that our data availability policy requires that all numerical data underlying display items are included with the submission, and you will need to provide this before we can formally accept your manuscript, if not already present. requires that all numerical data underlying display items are included with the submission, and you will need to provide this before we can formally accept your manuscript, if not already present. requires that all numerical data underlying display items are included with the submission, and you will need to provide this before we can formally accept your manuscript, if not already present. requires that all numerical data underlying display items are included with the submission, and you will need to provide this before we can formally accept your manuscript, if not already present.

**Press Queries**

If you or your institution will be preparing press materials for this manuscript, or if you need to know your paper's publication date for media purposes, please inform the journal staff as soon as possible so that your submission can be scheduled accordingly. Your manuscript will remain under a strict press embargo until the publication date and time. This means an early version of your manuscript will not be published ahead of your final version. PLOS Genetics may also choose to issue a press release for your article. If there's anything the journal should know or you'd like more information, please get in touch via plosgenetics@plos.org....

---

## [Editor Report · Acceptance letter]

PGENETICS-D-25-00434R2

Examining the association between fetal *HLA-C*, maternal , maternal , maternal , maternal *KIR* haplotypes and birth weighthaplotypes and birth weighthaplotypes and birth weighthaplotypes and birth weight

Dear Dr Decina,

We are pleased to inform you that your manuscript entitled "Examining the association between fetal *HLA-C*, maternal , maternal , maternal , maternal *KIR* haplotypes and birth weight" has been formally accepted for publication in PLOS Genetics! Your manuscript is now with our production department and you will be notified of the publication date in due course.haplotypes and birth weight" has been formally accepted for publication in PLOS Genetics! Your manuscript is now with our production department and you will be notified of the publication date in due course.haplotypes and birth weight" has been formally accepted for publication in PLOS Genetics! Your manuscript is now with our production department and you will be notified of the publication date in due course.haplotypes and birth weight" has been formally accepted for publication in PLOS Genetics! Your manuscript is now with our production department and you will be notified of the publication date in due course.

With kind regards,

Anita Estes

PLOS Genetics

On behalf of:
